# Strain design optimization using reinforcement learning

**Maryam Sabzevari**[1]*, **Sandor Szedmak**[1], **Merja Penttilä**[2], **Paula Jouhten**[2¤], **Juho Rousu**[1]

**1** Department of Computer Science, Aalto University, Espoo, Finland, **2** VTT Technical Research Centre of Finland Ltd, Espoo, Finland

¤ Current address: Department of Bioproducts and Biosystems, School of Chemical Engineering, Espoo, Finland

* maryam.sabzevari@aalto.fi, sabzevary.maryam@gmail.com

**Data Availability Statement:** All relevant data are within the manuscript and its Supporting Information files. An open source implementation of the algorithm is provided at https://github.com/maryamsabzevari-ai/strain-optimization.

## Abstract

Engineered microbial cells present a sustainable alternative to fossil-based synthesis of chemicals and fuels. Cellular synthesis routes are readily assembled and introduced into microbial strains using state-of-the-art synthetic biology tools. However, the optimization of the strains required to reach industrially feasible production levels is far less efficient. It typically relies on trial-and-error leading into high uncertainty in total duration and cost. New techniques that can cope with the complexity and limited mechanistic knowledge of the cellular regulation are called for guiding the strain optimization.

In this paper, we put forward a multi-agent reinforcement learning (MARL) approach that learns from experiments to tune the metabolic enzyme levels so that the production is improved. Our method is model-free and does not assume prior knowledge of the microbe's metabolic network or its regulation. The multi-agent approach is well-suited to make use of parallel experiments such as multi-well plates commonly used for screening microbial strains.

We demonstrate the method's capabilities using the genome-scale kinetic model of *Escherichia coli*, k-ecoli457, as a surrogate for an *in vivo* cell behaviour in cultivation experiments. We investigate the method's performance relevant for practical applicability in strain engineering i.e. the speed of convergence towards the optimum response, noise tolerance, and the statistical stability of the solutions found. We further evaluate the proposed MARL approach in improving L-tryptophan production by yeast *Saccharomyces cerevisiae*, using publicly available experimental data on the performance of a combinatorial strain library.

Overall, our results show that multi-agent reinforcement learning is a promising approach for guiding the strain optimization beyond mechanistic knowledge, with the goal of faster and more reliably obtaining industrially attractive production levels.

## Author summary

Engineered microbial cells offer a sustainable alternative solution to chemical production from fossil resources. However, to make the chemical production using microbial cells

**Funding:** JR was supported by grants from The Finnish Innovation Fund SITRA (www.sitra.fi), grant number 381202 and Academy of Finland (www.aka.fi) grant number 310107. PJ was supported by grants from Academy of Finland (https://www.aka.fi/en/) with grant numbers 310514, 314125, 335783, 352410 and 352412. MP was supported by grant from Jenny and Antti Wihuri Foundation (wihurinrahasto.fi). The funders had no role in study design, data collection and analysis, decision to publish, or preparation of the manuscript.

**Competing interests:** The authors have declared that no competing interests exist.

economically feasible, they need to be substantially optimized. Due to the biological complexity, this optimization to reach sufficiently high production is typically a costly trial and error process.

This paper presents an Artificial Intelligence (AI) approach to guide this task. Our tool learns a model from previous experiments and uses the model to suggest improvements to the engineering design, until a satisfactory production performance is reached. This paper evaluates the behaviour of the proposed AI method from several angles, including the amount of experiments needed, the tolerance to noise as well as the stability of the proposed designs.

This is a *PLOS Computational Biology* Methods paper.

## Introduction

Engineered microbial cells present a sustainable alternative to fossil-based chemical and fuel production [1]. Assembling and introducing the production routes into microbial strains is enabled by the state-of-the-art synthetic biology tools. However, to improve the initial laboratory demonstrated production to industrially feasible level requires substantial optimization of the cells. This strain optimization is typically challenging since the complex and insufficiently known cellular regulation has to be overcome to divert resources to production. Predictive models have been sought for guidance of the strain optimization process. Genome-scale metabolic models have already shown useful. Several strain design approaches using genome-scale metabolic models have been derived [2–4] but they are inherently limited to accounting for mass conservation and thermodynamic constraints to metabolism. They cannot account for instance kinetic constraints, allosteric regulation and regulatory interactions beyond metabolism, all playing important roles in metabolic flux distribution in cells.

Due to the limitations in the predictive models of cellular regulation the optimization of microbial cells commonly follows the Design, Build, Test, Learn cycle (DBTL). In this cycle, the strain is designed (D), built in a laboratory (B), measured, and tested (T) to learn (L) a model on the current strain to be exploited for the next cycle design phase (D) again. While efficient engineering solutions already exist for the build and testing phases, the design and learn phases of the DBTL cycle relies still strongly on manual evaluation by domain experts. This hinders the development of new industrially relevant productions strains [5].

In recent years, artificial intelligence and machine learning have emerged as approaches to facilitate the microbial strain design by being able to account for cellular regulation beyond established mechanistic knowledge [6–12]. For example, Hamerirad *et al.* (2019) [6] propose a Bayesian optimization approach using Gaussian Processes to tune the gene expression levels of three promoter of the lycopene production pathway of *E. coli* to maximize lycopene production. In another recent study a method called ART (Automated Recommendation Tool) [7], was proposed which leverages on a Bayesian ensemble approach to learn a mapping from the omics profiles of the strains to the outputs of interest.

In this study we propose a Reinforcement Learning (RL) [13] approach for automating the design and learn phases of the DBTL cycle for strain optimization. Our method facilitates the strain optimization by analysing the response (e.g. product yield) from the previous rounds

and by suggesting most promising modifications to the strain for the next round. Our method is a model-free RL approach [13–15], that is, it does not rely on prior knowledge on the structure and dynamics of the metabolic system. Instead, the method learns to predict the responses of given strain modifications and based on the predictions, recommends the most promising strain modifications to try out next. We further propose a Multi-Agent RL (MARL) approach [16], which is a good match of the parallel experimentation in the laboratory, such as using multi-well plates or multiple bioreactors running simultaneously [17–19].

In our case, we cannot assume a stationary model which allows to estimate the value function in a reliable way, therefore, in the implementation of the model-free RL our learning approach closely follows the scheme of the multi-armed bandit problem with associative tasks, also called as contextual bandit [13, 20]. In the contextual bandit framework only the current value of the reward is used with the combination of the history of the context, the earlier observed states and the modifications executed until the current observation. To support these kind of learning approaches several theoretical models [20, 21] and practical realizations [22, 23], have already been developed.

The main contributions of this paper are summarized in the following:

- We present a model-free reinforcement learning method that proposes modifications to enzyme levels based on the previous experiments.

- A Multi-agent extension is presented that is able to take advantage of parallel experiments, such as screening using multi-well plates or other parallelized cultivation systems.

- A comprehensive empirical evaluation is presented on the genome-scale kinetic model of *E. coli* (k-ecoli457), evaluating the sample complexity, noise tolerance, and stability of the designs.

- The proposed MARL method is assessed in improving L-tryptophan production with yeast *Saccharomyces cerevisiae*, using publicly available experimental data on performance of a library of yeast strains.

- An open source implementation of the algorithm is provided at https://github.com/maryamsabzevari-ai/strain-optimization

## Materials and methods

### Reinforcement learning for optimising strain designs

Reinforcement learning (RL) refers to a goal-oriented learning approach via interacting with an environment. In RL, an agent $x$ aims to maximize a reward $r$ by making iterative decisions, or actions **a**, based on the observations of the states **s** of the environment. The decision are guided by a policy $\pi$ that gives suitable actions in a given state.

Our overall DBTL cycle using multi-agent reinforcement learning (MARL) is shown in Fig 1. We assume that each round of the cycle corresponds to a set of cultivations run in parallel, and for each cycle a new set of strains is generated, based on the recommendations of the agents. In each round the system is observed in a pseudo steady-state (e.g. corresponding to cells in exponential growth phase), which is estimated by the observable variables. The observable variables correspond here to metabolite concentration and enzyme expression level variables, whose values are assessed at appropriate state of the cultivations. Actions correspond to genetic engineering steps that increase or decrease the levels of the metabolic enzymes that have been chosen as the controllable variables. Rewards correspond to improvement of a target

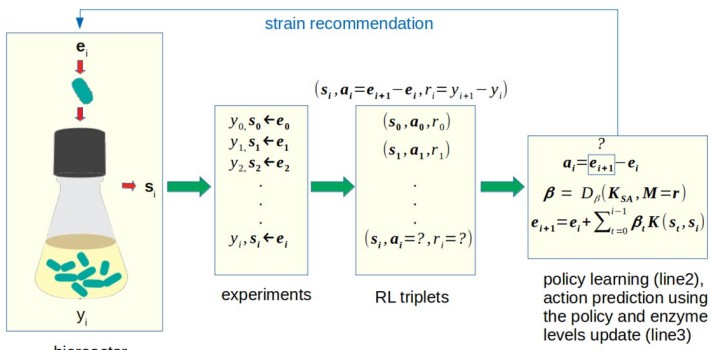

**Fig 1. Strain design optimization loop using reinforcement learning.** Enzyme levels corresponding to the strain $i$ are denoted as $\mathbf{e}_i$, and $y_i$ and $\mathbf{s}_i$, correspond to the response (used in reward) and output concentrations (used as state), respectively. The action ($\mathbf{a}_i$), corresponding to the difference of the enzyme levels in the two consecutive iterations, is given by the policy learned with MMR.

variable such as the product yield or a product of the specific production and growth rates. The policy corresponds to mapping the state of the system (estimated by the observations) to a set of changes of the enzyme levels.

The components of the framework are summarized below:

**Actions** $\mathbf{a} \in A = \mathbb{R}^{n_a}$, real valued vectors containing the changes of the enzyme levels (dimension $n_a$)

**States** $\mathbf{s} \in S = \mathbb{R}^{n_s}$, vectors of the steady state concentrations of metabolites and enzymes (dimension $n_s$).

**Rewards** $r_t \in \mathbb{R}$: the change of the target variable $y$ (e.g. product yield) between two consequtive rounds $r_t = y_t - y_{t-1}$

**Policy** $\pi : S \mapsto A$: mapping from the states to actions, learned from data

A heatmap illustrating the changes of the enzyme levels throughout the optimization process for a particular product (*succinic acid*), using the proposed MARL algorithm is presented in Fig 2.

## Learning algorithm

In constructing the learning algorithm, we aim to find an optimal policy $\pi^*$ which maps the observed states to the promising actions. The basic task for an agent at round $t$ is to find an action $\mathbf{a}_{t-1}$, to likely improve the reward $r_{t-1}$, based on the observed state $\mathbf{s}_{t-1}$ of the system.

To implement this, we maintain a history $\mathcal{H}_t = \{\mathbf{S}_t, \mathbf{A}_t, \mathbf{R}_t\}$, of the state-action-reward triples $(\mathbf{s}, \mathbf{a}, r)$ obtained in the previous rounds, stored in matrices $\mathbf{S}_t$, $\mathbf{A}_t$ and $\mathbf{R}_t$, which will be used to learn the policy $\pi_t$ for the next round.

To realize the prediction of next action, the policy function, we construct an optimization problem which aims to learn the relationship between the states and the actions.

We apply the Maximum Margin Regression (MMR) framework [24], which builds on the Support Vector Machine (SVM) and is able to predict not only binary outputs but arbitrary responses given by vectors of a Hilbert space. As a heritage from the SVM, the MMR can also use kernel represented input vectors to express potentially nonlinear relations as well. Since the MMR allows to learn vector outputs with internal structure, therefore the potential

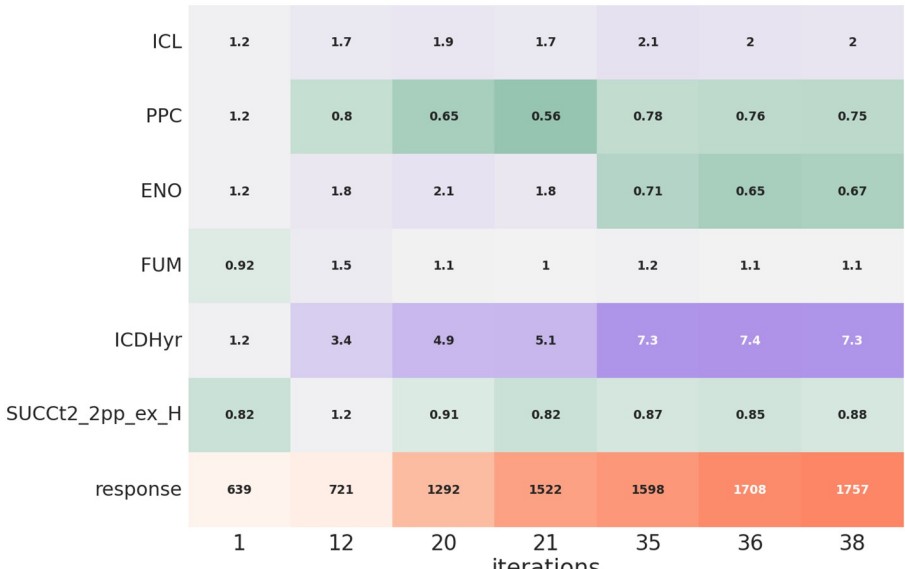

**Fig 2. Illustration of strain design optimization using the MARL approach for *succinic acid*.** The first 6 rows correspond to the enzyme levels and in the last row the response (product exchange flux * growth) is presented. Each column represents the enzyme levels design and the corresponding response which has been found in the iteration mentioned at the bottom.

interdependence between the output components can be expressed. The MMR is the simplest known approach to implement these kind of regression problems. The alternatives, e.g. Structured SVM, see several implementations in [25], and the Input Output Kernel Regression, [26], have significantly higher computational complexity, thus their use in an iterative, Reinforcement Learning framework could be too expensive. The computational complexity of solving the optimization problem realizing an MMR based learner is the same as the complexity of the SVM applied to binary classification.

MMR learns the policy through a linear operator $\mathbf{W}$: $H_S \to H_A$, where $H_S$ and $H_A$ are feature spaces (Hilbert spaces) corresponding to the states and actions, respectively. Feature maps $\boldsymbol{\phi}$: $S \to H_S$, and $\boldsymbol{\psi}$: $A \to H_A$ are used to map the states and actions to their respective feature spaces.

The predicted action by the learned policy in state $\mathbf{s}$ is given by

$$\pi(\mathbf{s}) = \arg\max_{\mathbf{a} \in A} \langle \boldsymbol{\psi}(\mathbf{a}), \mathbf{W}\boldsymbol{\phi}(\mathbf{s}) \rangle,$$

where the inner product $\langle \boldsymbol{\psi}(\mathbf{a}), \mathbf{W}\,\boldsymbol{\phi}(\mathbf{s}) \rangle$ can be interpreted as a predicted reward of action $\mathbf{a}$ in state $\mathbf{s}$. When the feature map $\psi$ is a surjection, the above maximization has the alternative solution $\pi(\mathbf{s}) = \psi^{-1}(\mathbf{W}\,\boldsymbol{\phi}(\mathbf{s}))$, where $\psi^{-1}$: $H_A \to A$ is the pre-image of $\psi$.

The learning is guided by a constraint calling for all triplets $(\mathbf{s}_i, \mathbf{a}_i, r_i)$ in training data the predicted reward to be lower bounded by a margin, a strictly monotonically increasing function $M : \mathbb{R} \to$ of the observed reward:

$$\langle \boldsymbol{\psi}(\mathbf{a}_i), \mathbf{W}\boldsymbol{\phi}(\mathbf{s}_i) \rangle \geq M(r_i)$$

The type of constraint encourages the model to use the triplets with a high reward as support vectors and thus improves the model's performance in the high-reward regime.

The full optimization problem is formulated in the following way:

$$\min \quad \frac{1}{2}\|\mathbf{W}\|_F^2 + \mathbf{C}_\xi \sum_i \xi_i$$

$$\text{w.r.t.} \quad \mathbf{W} \ (linear \ operator) : S \to A,$$

$$\boldsymbol{\xi} \in \mathbb{R}^{t-1} \tag{1}$$

$$\text{s.t.} \quad \langle \mathbf{a}_i, \mathbf{W}\boldsymbol{\phi}(\mathbf{s}_i)\rangle \geq M(r_i) - \xi_i,$$

$$\xi_i \geq 0, \ \ i = 1, \ldots, t-1.$$

The objective minimizes a sum of a regularizer (squared Frobenius norm of $\mathbf{W}$) and an error term consisting of sum of slack variables for the training data. The slacks allow outliers, triplets that fail to achieve a required margin. For the margin, we applied log-transformed rewards: $M(r) = log_2(r + 1)$. The hyper-parameter $C_\xi$ controls the trade-off between regularization and error.

Like with SVM, efficient non-linear modelling can be achieved by the use of non-linear kernel functions, which circumvent the need for constructing the potentially high-dimensional feature maps explicitly. We apply Gaussian kernels $\mathbf{K}_S(\mathbf{s}, \mathbf{s}') = \exp(\|\mathbf{s} - \mathbf{s}'\|^2/\sigma_S^2)$ and $\mathbf{K}_A(\mathbf{a}, \mathbf{a}') = \exp(\|\mathbf{a} - \mathbf{a}'\|^2/\sigma_A^2)$ for the state and action spaces, as well as a joint kernel $K_{SA}(\mathbf{s}, \mathbf{a}; \mathbf{s}', \mathbf{a}') = \mathbf{K}_S(\mathbf{s}, \mathbf{s}')\mathbf{K}_A(\mathbf{a}, \mathbf{a}')$.

Using the above kernels, a Lagrangian dual problem of (1) can be derived, where the predicted action in state $\mathbf{s}$ is expressed as

$$\pi(\mathbf{s}) = \sum_i \beta_i \mathbf{a}_i \mathbf{K}_S(\mathbf{s}, \mathbf{s}_i), \tag{2}$$

as an approximate pre-image solution of Gaussian prediction over actions.

The solution to the above problem is computed by solving the the dual optimization problem [24]:

$$\boldsymbol{\beta}^* = D_\beta(\mathbf{K}_{SA}, \mathbf{M}, C_\xi) = \arg \begin{cases} \min & \frac{1}{2}\boldsymbol{\beta}'\mathbf{K}_{SA}\boldsymbol{\beta} - \mathbf{M}^T\boldsymbol{\beta} \\ \text{w.r.t.} & \boldsymbol{\beta} \in \mathbb{R}^{t-1}, \\ \text{s.t.} & \mathbf{0} \leq \boldsymbol{\beta} \leq C_\xi \mathbf{1} \end{cases} \tag{3}$$

where $\mathbf{K}_{S\,A} = \mathbf{K}_S \circ \mathbf{K}_A$ is the element-wise product of the kernels matrices of the states and the actions.

## Multi-agent algorithm

The learning algorithm described above learns sequentially, from one experiment per DBTL round. However, modern wet-lab arrangements typically allow parallel execution of experiments either by using multi-well plates or micro-bioreactors. In general, the parallel execution of experiments have a lower overhead, due to the lower human labour cost. However, from machine learning point of view, fully parallel execution does not yield optimal results as most learning happens between the DBTL rounds. MARL employs several co-operating agents that learn from each other and together explore the solution space.

MARL setups can be generally defined using centralized or decentralized schemes in either/ both of training and execution phases [16]. A centralized training scheme enables the agents to

learn from each other, which can improve sample efficiency [27, 28], critical in the applications where environment interaction (e.g. Build and Test phases in DBTL) is expensive. However, decentralized learning allows the agents to adapt to specific regimes of the state space, and thus, obtains better overall performance through diversity among the agents [29]. Various structures for mixing centralized and decentralized schemes have been studied, due to their diverse advantages which can be helpful in complex real-world tasks [30–32].

In the following, we show how the strain optimization task can be tackled with MARL approach, where several agents provide strain recommendations to be executed in parallel within a single DBTL round.

Our MARL algorithm (Algorithm 1) uses a mixture of centralized and decentralized training, realized by grouping the agents so that within group $g$ the agents are learning a joint policy $\pi_g$ based on the history $H_g$ of the group (centralized part). Across groups the agents learn independently, which allows each agent group to specialize to a specific region of the state space (decentralized part). All agents share the same actions space $A$ and state space $S$. However, each agents occupies its own state at any given iteration.

The algorithm begins with an initialization phase where an initial data set is generated as a warm-up to the MARL algorithm and initial enzyme levels are chosen (see section below for details).

In each iteration of the MARL algorithm $t$, an agent $x_i$ in group $g$ observes the state $\mathbf{s}_t^i$, and executes an action $\mathbf{a}_t^i$ based on the group policy $\pi_g$, which transitions the system to the next state $\mathbf{s}_{t+1}^i$, and the reward $r_t^i$ is obtained.

The parameters $\boldsymbol{\beta}_g$ of the joint policy $\pi_g$ for each group $g$, are updated in each iteration using the Eq (3), using the groups history $\mathcal{H}_g = \{\mathbf{S_t^g}, \mathbf{A_t^g}, \mathbf{R_t^g}\}$ as the training data, where the matrices of states, actions and rewards are built by concatenating the individual matrices of the agents in the group.

To avoid the the agents of each group to become too homogeneous, the actions are perturbed to increase their diversity before their execution (see below for details). In addition, every few iterations, the worst performing agent is replaced with a copy of randomly selected other agent.

**Initialization.**   Since the algorithm uses the experiments that have been collected previously as training examples, an empty memory in the beginning produces arbitrary actions that can be avoided by including the warming up phase to initiate the memory. The warm-up period and controllable factors initialization is implemented as described below:

- **Warm-up**: In this study, as in [7], we have used Latin hyper-cube sampling (LHS) [33] to ensure having diverse initial experiments representing the input space variability. In LHS, each dimension's range is divided into non-overlapping, equally probable intervals. Hence the input space is divided into equiprobable subspaces. Each sample should be taken from a hyper-row and hyper-column, which does not contain any of the previous samples. We build up a history of the data obtained from $\tau$ performed interactions with the environment for each agent $x_i$, which will be later used to predict the future actions.

- **Choosing initial enzyme levels**: For the first round ($t = \tau + 1$), the enzyme levels of the first agent are initiated using the median of the the enzyme levels corresponding to the upper quantile of the observed responses in the warm-up. Enzyme levels of the other agents are initialized randomly to ensure exploration of different sub-spaces.

**Algorithm 1** Multi agent Max-Margin Strain Design Optimization Algorithm
**Input:** $\tau$: *warm-up iterations*, X: *agents*, $G = \{g_j\}_{j=1}^{n_g}$: *agent groups*, S: *state space*, A: *action space*

```
Output: e: enzyme levels
```
$(\{H_\tau^j\}_{j=1}^{n_g}, \{\mathbf{s}_\tau^i\}_{i=1}^{n_x}, \{\mathbf{e}_\tau^i)\}_{i=1}^{n_x}) = \mathbf{Initialize}(S, A, \tau)$
$t = \tau + 1$
```
Repeat:
  For j = 1...n_g: # loop over the groups of agents
```
 $(\mathbf{s}_t^i, y_t^i) \leftarrow \mathcal{E}(\mathbf{e}_t^i) \ \forall x_i \in g_j$ # *each agent makes an experiment*
 $H_t^j = H_t^j \cup (\mathbf{s}_t^i, \mathbf{a}_t^i, r_t^i = y_t^i - y_{t-1}^i) \ \forall x_i \in g_j$ # *update group history*
 $\pi_j = \mathbf{MMR}(H_t^j)$ # *update group policy using MMR* (3)
 $\mathbf{a}_t^i = \pi_j(\mathbf{s}_t^i) \ \forall \ x_i \in g_j$ # *predict actions:*
 $[\tilde{\mathbf{a}}_t^i]_{i \in g_j} = \mathbf{Perturb\_agents}([\mathbf{a}_t^i]_{x_i \in g_j})$ # *diversify the actions of the group*
 $\mathbf{e}_{t+1}^i = \mathbf{e}_t^i + \tilde{\mathbf{a}}_t^i, \ \forall \ \mathbf{x}_i \in X$ # *compute new enzyme levels for each agent*
```
  If t mod k = 0 # every k iterations
```
 $i_{worst} = \mathbf{argmin}_i \ \mathbf{median}_t \ y_t^i$ # *worst agent*
 $\mathbf{e}_t^{i_{worst}} = \mathbf{e}_t^{i_{random}}$ # *substitute with a random agent*
```
  t = t+1
Until convergence
```

**Perturbation scheme.** One of the main purpose of applying multi-agents in the learning procedure is to increase the explorative capacity of the learner. This requires an algorithm in which the divergence of the actions among the agents is sufficiently large. The basic idea is to increase the volume of the parallelotope spanned by the action vectors. That volume can be expressed by the determinant of the Gram matrix, $[\mathbf{G}_{ij}]_{i,j=1}^{N} = \langle \mathbf{a}_i . \mathbf{a}_j \rangle$ consisting of the inner product between the action vectors, and the greater determinant implies greater volume. The maximum volume corresponds to case where those actions are orthogonal, and the minimum is achieved when they point into the same direction. In the presented algorithm the perturbation vectors for each group ($g_j$) of the agents are created by the Gram-Schmidt orthogonalization of the actions, then those perturbations are scaled by a fixed perturbation coefficient ($c = 0.8$) and added to the original action vectors. Finally the length of the actions are restored to the original one. It can be proved that the volume corresponding to the Gram matrix of the actions increases by this algorithm. A summary of this procedure in presented in algorithm 2.

**Algorithm 2** Perturb_agents
```
Input: actions: a¹, ..., a^sz, perturbation coefficient c
Output: Perturbed actions: ã¹,...,ã^sz
```
$[\mathbf{v}^i]_{i=1}^{sz} = \mathbf{Gram-Schmidt}([\mathbf{a}^i]_{i=1}^{sz})$
```
For i = 1...sz:
```
 $\tilde{\mathbf{a}}^i = (1-c)\mathbf{a}^i + c\mathbf{v}^i$ # *action perturbation*
 $\tilde{\mathbf{a}}^i = \frac{\|\mathbf{a}^i\|\tilde{\mathbf{a}}^i}{\|\tilde{\mathbf{a}}^i\|}$ # *Normalizing to the original length*

## Enzyme level tuning in k-ecoli457

In this work, we have used simulations from k-ecoli457 [34] genome-scale kinetic model of *E. coli* metabolism as a surrogate for wet-lab experimentation. The k-ecoli457 model has been validated on fluxomic data corresponding to wild-type and 25 mutant strains (21 genetically perturbed strains with glucose as the carbon substrate and four with different carbon substrates under aerobic conditions) growing under different conditions on various substrates. As the authors of this model reported in their study, the Pearson correlation coefficient between the experimental data and the product yield predicted using this model for 320 engineered strains is notably higher than using flux balance analysis [2] or minimization of metabolic adjustment [35] for predicting the product yield. This comprehensive kinetic model with 457 reactions, 337 metabolites, and 295 substrate-level regulatory interactions seems to be an appropriate candidate for strain optimization simulation.

In k-ecoli457 model, change of the enzyme levels is implemented by varying the total pool of the normalized enzymes from wild-type ($\hat{e}_{tot} = 1$) down to 10-fold down-regulation ($0.1 \leq \hat{e}_{tot} < 1$) and up to 10-fold up-regulation ($1 < \hat{e}_{tot} \leq 10$).

For the purpose of the MARL algorithm, enzyme levels are transformed via $f = \frac{-1+\hat{e}_{tot}}{\hat{e}_{tot}+1}$, which induces nearly symmetric ranges for of $(-1, 0)$, $(1, 0.82]$, $-1$ and $0$ for down-regulation, up-regulation, knock-out and no regulation, respectively. Inverse of the transformation function $f$ ($f^{-1} = \frac{-y-1)}{y-1}$), is used to transform back the enzyme level for interacting with the simulator (environment).

## Choice of controllable variables and products

In Table 1, the list of the perturbed enzymes for three investigated products *acetate*, *ethanol* and *succinic acid* is presented, chosen according to [34] and [36]. The strain designs for improved production are sought for using a target quantity [production x growth] similar to [37], and referred to as response from here on, to avoid solutions severely compromising the viability of cells. The response for each product on the carbon source glucose is computed as $rr_{target\text{-}reaction}{}^{*} rr_{growth\text{-}reaction}$, where the first term denotes the specific production rate of the target compound and the second term denotes the growth. All the simulations were performed under aerobic conditions with glucose as a non-depleting carbon source (i.e. external concentration independent of utilization) [34]. Enzyme level perturbation has been performed by varying (increasing or decreasing) the enzyme levels within the range [0.1, 10], in which [0.1, 1] corresponds to 10-fold down-regulation up to wild-type level range and [1, 10] corresponds to wild-type level up to 10-fold up-regulation (more information in [34]). We have discarded gene deletion due to causing abrupt (non-continuous) changes in the yield space. The modified enzyme levels set in each iteration of the algorithm is queried in the simulator, to check for the obtained response improvement over the best found response yet. In addition of the response improvement, we are interested to evaluate the stability of the obtained response in the response region. This is mainly crucial, as in the real-world scenario the error-prone measurements might land the experiment in a close neighborhood of the computed enzyme levels rather than in a exact point of *in vivo* optimality in the enzyme level space.

In this study, we assume running 20 parallel experiments (4 technical replicates times 5 biological replicates) and 40 sequential iterations search performed in the enzyme level space.

## Compared strain optimization methods

To asses the proposed MARL method for strain design optimization, we carried out a comparison against Bayesian Optimization with Gaussian process as surrogate function (BO-GP) [38, 39] and also random search (RAND), which has been shown theoretically and empirically to be more efficient than grid search for hyper-parameter optimization [40].

**Table 1. List of the investigated products of interest and the corresponding enzymes whose levels were optimized, selected based on [34] and [36].**

| product | target reaction | perturbed enzymes |
|---|---|---|
| succinic acid [36] | SUCOS | ICL, PPC, ENO, FUM, ICDHyr, SUCCt2_2pp_ex_H |
| acetate [34] | EX_ac(e) | RPI, PFL, GLCptspp_ex_exi, HSK, THRS, IPPS, IPMD, OMCDC, IPPMIb, IPPMIa, HSDy, ASPK, LDH_D |
| ethanol [34] | PDH | PFL, FRD2, RPI, GLCptspp_ex_exi, HSK, LDH_D, ASPK, THRS, IPPS, IPMD, OMCDC, IPPMIb, IPPMIa |

For the all three algorithms, we have used the same examples for the warm up period. In the experiments, four parallel agents are considered by the MARL algorithm, and to match the setup, in the BO-GP and RAND experiments, four parallel experiments are executed. We have used an existing Bayesian Optimization implementation in SHERPA [41], for BO-GP.

All the reported experiments relate to the average of the best found response obtained by iteration 40—a point where the improvement of the response variable has slowed down for all methods—across four agents and five runs of varied seeds (resembling biological replicates).

## Results

We report results on evaluating of the proposed MARL method both in simulation data arising from the k-ecoli457 model, as well as on experimental data arising from [8].

In the simulation experiments, the MARL approach is used to optimize the k-ecoli457 model's production of *acetate*, *ethanol* or *succinic acid* multiplied by growth (i.e. response). All the reported responses are relative to their corresponding wild-type cases, that is, improvement obtained for each product with respect to its corresponding wild-type case. Besides the response optimization, we also study the stability of the solutions and the noise tolerance of algorithms.

For the evaluation of the proposed method on experimental data, we used a combinatorial yeast *S. cerevisiae* strain library provided in [8]. In this problem, the aim was improving L-tryptophan production by optimizing the expression levels of five genes *PCK1*, *TAL1*, *TKL1*, *CDC19* and *PFK1*.

### Response improvement

The response improvements obtained for the aforementioned products in k-ecoli457, using the studied algorithms, MARL, BO-GP and RAND, are presented in this section.

In Fig 3, we have illustrated the median (solid lines) and 25th to 75th percentiles (shaded areas) of the obtained best found response improvement of the MARL approach with respect to BO-GP and RAND throughout the iterations, for the three products, *acetate*, *ethanol* and *succinic acid*. MARL approach reached in earlier iterations than BO-GP and RAND improvement in both *acetate* and *succinic acid*, and it outperformed the two other approaches in *ethanol* as early as iteration 13.

### Statistical stability of the optimal strain designs

Here we set out to to study how sensitive the final strain design obtained by the algorithm is to the situation that the actual enzyme levels realized are different from the designed ones. For quantifying the stability of a particular strain design, we computed the variation in the response (product exchange flux * growth) of the final strain design within a neighborhood generated by small perturbations of strains. Specifically, we sampled randomly within 0.5 fold change of enzyme levels of the final strain design and checked the target response in each sampled point. We used as a measure of strain stability the relative standard deviation (or coefficient of variation) $RSD = \frac{\sigma}{\mu}$ of the yields, where $\sigma$ and $\mu$ denote standard deviation and mean of the neighbors' responses, respectively.

We ran MARL for 40 iterations, and considered the strain design with the best response within those iterations as the final strain design. We computed in total 20 trajectories per product, consisting of four agents, and five repeated trials per agent. We used neighborhood size of ten for computing the stability measure (RSD). RL appears to converge to more stable strain designs than RAND meaning that the close neighbourhood of the recommended strains by

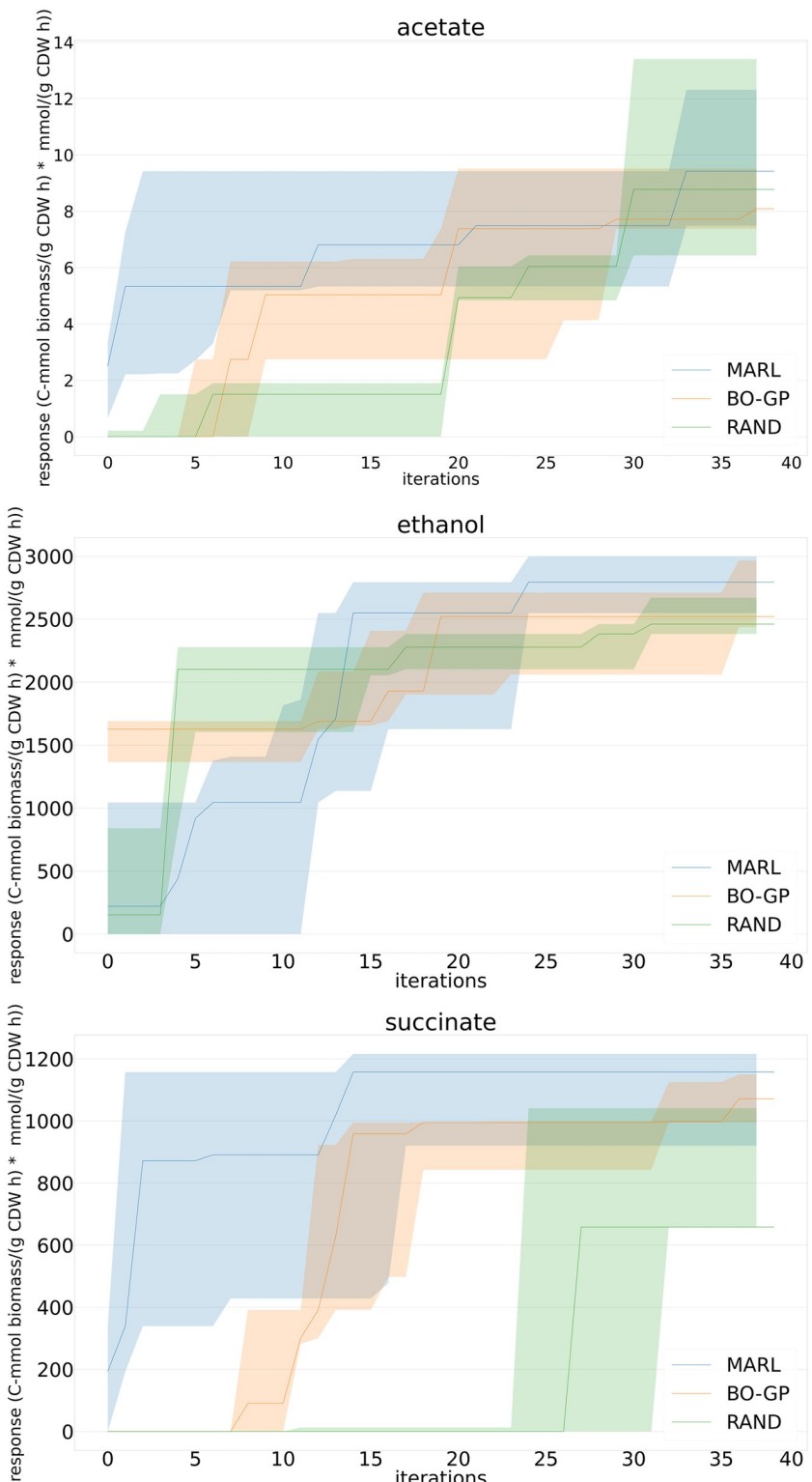

**Fig 3. The development of the median (solid lines) and 25th to the 75th percentile (shaded areas) of the response (growth∗production, C-mmol biomass/(g CDW h)∗mmol/(g CDW h)) in *k-ecoli457* model using MARL, BO-GP and RAND, in *acetate* (top), *ethanol* (middle) and *succinic acid* (bottom) production.**

**Table 2. Mean and standard deviations of the strain stability measure (RSD) obtained with algorithms.** For each product and each method mean and standard deviation among 20 computed RSDs are shown.

| product | method | mean±std |
|---|---|---|
| acetate | RL | 30.49±16.21 |
| | BO-GP | 45.86±24.44 |
| | RAND | 49.67±14.37 |
| ethanol | RL | 23.59±21.18 |
| | BO-GP | 71.33±55.58 |
| | RAND | 51.36±28.38 |
| succinic acid | RL | 34.11±25.31 |
| | BO-GP | 33.47±22.53 |
| | RAND | 57.62±31.87 |

MARL are less likely to result in a drastic different response (Table 2). Compared to BO-GP, more stable strain designs are found by MARL for *acetate* and *ethanol*, and for *succinic acid* the stability of the strain designs obtained with BO-GP and MARL are comparable. However, the final *succinic acid* response with BO-GP is significantly lower than the response obtained with either MARL or BO-GP, which lessens the worth of the good stability.

## Production improvement in the presence of noise

Using the strain designs in practice is subject to noise arising from different sources (measurement errors, biological variability, inaccuracy of implementing enzyme level changes). Since the k-ecoli457 model is deterministic, it lacks all of these sources of variability and, consequently, might make the algorithm's performance look unrealistically good compared to the real-life situation. To evaluate the robustness of the proposed MARL strategy under noise, we performed simulations related to MARL approach in a setup where noise was added to the input or output variables of the k-ecoli457 model. We experimented with noise levels of $m$% ($m \in [10, 20, 30]$) noise in either states, actions or response (product exchange flux $^*$ growth).

To simulate noisy states (concentrations of metabolites and enzymes in k-ecoli457 output), the original state vector was added to a Gaussian noise signal denoted as $X \sim N(\mu, \sigma^2)$ which is normally distributed with mean $\mu = 0$ and standard deviation equal to $m$% of the median of the values in the state vector. Similarly, the action vector (changes on the enzyme levels input to k-ecoli457) is summed up with a Gaussian noise signal with mean $\mu = 0$ and standard deviation equal to $m$% of the minimum absolute value in the action vector. The obtained growth-coupled production (response) is also perturbed by a Gaussian noise signal with mean $\mu = 0$ and standard deviation equal to $m$% of the original growth-coupled production. The average percentage of median max-response decrements at iteration 40 over three studied products was computed for the three investigated noise levels for the case when noise is added to the action (blue bars), response (orange bars), states (green bars) and also to the all three elements (pink bars), with respect to no noise scenario. In none of the simulations a significant adverse effect was observed, which reveals the robustness of the proposed MARL method in the presence of noisy measurements (Fig 4).

## Evaluation of strain design optimization using experimental data

In this section we report the evaluation of our proposed MARL approach on experimental data from a combinatorial yeast *S. cerevisiae* strain library provided by Zhang et al. (2020) [8] (Supplementary Data 3). The combinatorial strain library was aimed at improving L-

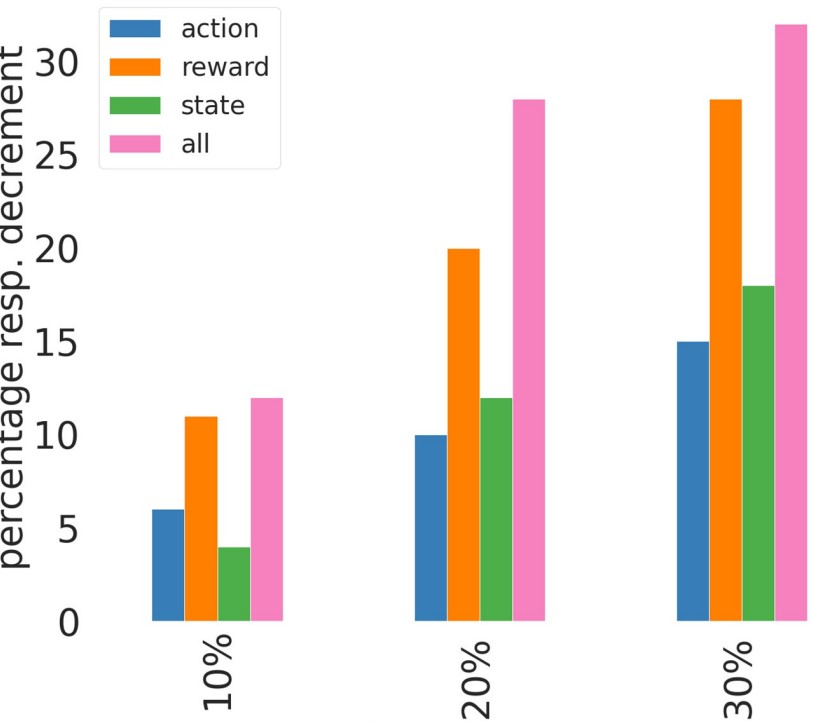

**Fig 4. Average percentage of median response decrements at iteration 40 over 3 studied products, with respect to the baseline with no noise.**

tryptophan production by optimizing the expression levels of five genes *PCK1*, *TAL1*, *TKL1*, *CDC19* and *PFK1*. As a proxy of L-tryptophan production a green-fluorescent protein (GFP) synthesis rate readout as a reporter of an engineered tryptophan biosensor had been determined. This comprehensive data set allowed us to evaluate designs proposed by the MARL algorithm, by selecting each time the closest matching design from the library.

To this end, the action vector (**a**) consists of 6 levels of genes strength coded as {0, 0.2, 0.4, 0.6, 0.8, 1}, respectively correspond to the lowest strength up to the highest of each gene. The state vector (**s**) consists of all the information available for each strain (the entire row for each strain of Supplementary Data 3 in [8]). The reward (*r*) is defined as the change of the GFP synthesis rate, which is the optimization objective in this problem.

The setting of the algorithm is kept as previously mentioned. At each iteration, for each recommendation given by the algorithm, we search for its nearest neighbor strain (element-wise subtraction for each coded genes strengths is less than 0.2) in the library to use it as the strain that has to be queried. In the case that no close strain is found, zeros is assigned to the GFP rate of that experiment. This assumption is not destructive in our experiments as we are looking at the best found GFP synthesis rate up to the current iteration, hence this choice will only miss the strains that can potentially produce higher GFP synthesis rates but are not provided in the available data set. We use this approach as a proxy for the lab experimentation.

We compare the proposed MARL algorithm with RAND and BO-GP as explained in the previous experiments. In Fig 5 we illustrate the average and 25th to 75th percentiles (across four agents and 20 times running the algorithms) of the best found GFP synthesis rates using MARL, BO-GP an RAND. Note that, for this experiment, we have not used the

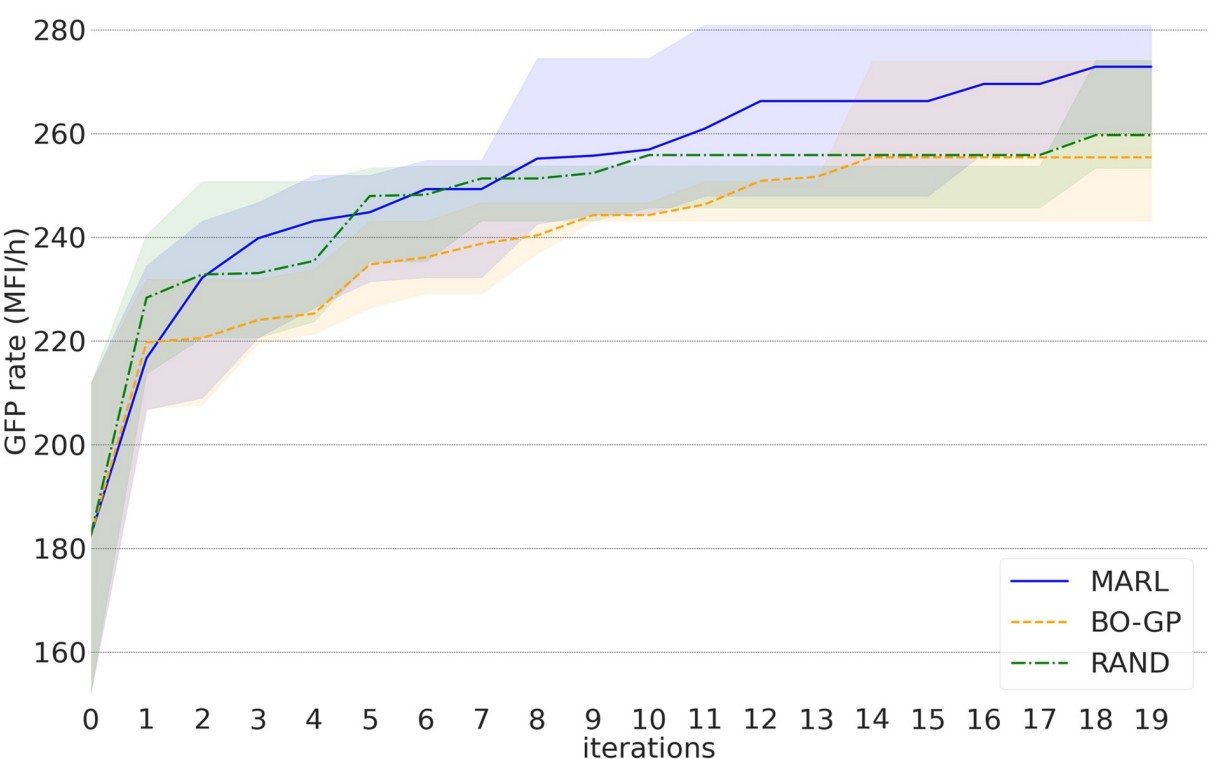

**Fig 5. Mean and 25th to 75th percentile (shaded areas) of the best GFP synthesis rate (MFI/h), using MARL, BO-GP and RAND.**

recommendation strains for training. However, to asses whether the algorithm finds the strains proposed by [8], we compare each recommendation with all the strains to monitor their corresponding GFP synthesis rates. As we can see that the proposed MARL approach outperforms BO-GP and RAND in terms of the average obtained GFP synthesis rate (MFI/h). Also we observe in this plot, MARL tends to find the highest GFP synthesis rates (blue shaded area).

## Discussion

In this paper we have put forward a MARL framework for strain design optimization. The proposed MARL algorithm is able to find metabolic enzyme levels that would optimize a given target such as desired compound production. MARL does not assume any stoichiometry or metabolic kinetics to be known but learns those dependencies and beyond between the given target and the enzyme levels from strain characterization data. Optimizing the enzyme levels has been found useful in improving production [42] and in wet-lab the enzyme levels can be adjusted by using e.g. different promoters, numbers of gene copies, or synthetic degrons. Though wide promoter libraries and other options for tuning the enzyme levels are available for many hosts, desired levels may not be achieved exactly. For this reason it is important that the designs are stable around the optimal enzyme levels. In our simulations using the k-ecoli457 model as a proxy for the wet-lab, our MARL approach in general showed better stability of the enzyme level designs upon small perturbations than the alternative BO-GP algorithm. The actual enzyme levels in engineered strains can be quantified by sampling cultivations and using targeted mass spectrometry based proteomics approaches. Neither

noise in the enzyme levels (i.e. input data) nor in the target (i.e. output data) substantially compromised the performance of our MARL algorithm, instead a gradual (linear) deterioration of obtained response levels was observed upon increasing noise levels.

We demonstrated the versatility of the algorithm by optimizing L-tryptophan production in *S. cerevisiae* using the combinatorial library data by Zhang et al. (2020) [8]. Our results show that MARL could optimize the L-tryptophan production (response variable being the GFP synthesis rate in response to L-tryptophan biosensor) effectively, achieving 95% of the optimum mean response within 12 iterations, while BO-GP algorithm not achieving this level within 19 iterations. Thus, MARL would be suitable for optimizing enzyme levels when integrated to an experimental DBTL cycle.

For the practical use of MARL integrated to an experimental DBTL cycle, we note that when the experiments were replaced with the k-ecoli457 model simulations, around 10-15 iterations of the MARL algorithm (i.e. corresponding to wetlab strain improvement and characterization iterations) were required to reach a median response that is within 10% of the optimum median response in case of improving *ethanol* or *succinic acid* production, while for improving *acetate* production more than thirty iteration were needed. This number depends, in part, on the number of enzymes that are being simultaneously optimized which here was between 6-13 enzymes. Focusing on a smaller set of enzymes is likely to lead to lower number of required samples to reach convergence, but at the same time, usually a lower optimum response will be exhibited. Another direction is to increase the parallelism by using more agents per iteration (i.e. corresponding to parallel modified strains in wetlab). This is also likely to decrease the number of iterations required, however, the cost per iteration will be higher, and the overall number of strain designs required will be higher.

MARL could also be used for optimizing metabolic gene expression levels. In that case, we expect that the amount of samples required by the algorithm is higher than in the enzyme level optimization, due to the more indirect link to metabolic fluxes. Similarly, response variables alternative to the product exchange flux by growth we restricted our attention to in the simulation studies or the biosensor's GFP reporter readout in the L-tryptophan case (i.e. Zhang et al. (2020) [8] are directly compatible with MARL.

Current limitation of our MARL method is that the strain design optimization is restricted to the given target enzymes though the number of target enzymes can be varied. However, target enzymes whose levels should be tuned for optimizing production have previously been successfully identified using a genome-scale metabolic model simulations [8]. If suitable prior knowledge is not available, we believe using genome-scale metabolic models for the target enzyme identification for MARL is a strong approach as it relies on the ultimate limits of metabolic states, mass conservation and thermodynamics. Genome-scale metabolic model simulations could also be used for proposing enzyme deletions leading to overproduction previously successfully predicted [43–45] before strain design optimization using MARL. However, the gene deletions are limited to the very few pathways that compete with production but are not essential for sufficient cell growth. Further development of MARL could also be using fluxes of modified strains predicted with genome-scale metabolic model simulations (with established methods such as e.g. MOMA [35], ROOM [46]) as an input for MARL together with experimental strain characterization data. Finally, optimizing the actual enzyme levels for maximizing production calls for a data driven approach like our MARL method due to the complexity of cellular regulation beyond known stoichiometric dependencies and kinetic constraints.

Overall, our results show that MARL is a promising approach for guiding the strain optimization beyond mechanistic knowledge, and has potential to contribute to advancing novel microbial strains reaching industrial production processes.

## Author Contributions

**Conceptualization:** Maryam Sabzevari, Sandor Szedmak, Merja Penttilä, Juho Rousu.

**Data curation:** Maryam Sabzevari.

**Formal analysis:** Maryam Sabzevari, Sandor Szedmak.

**Funding acquisition:** Merja Penttilä, Juho Rousu.

**Investigation:** Maryam Sabzevari.

**Methodology:** Maryam Sabzevari, Sandor Szedmak, Paula Jouhten, Juho Rousu.

**Project administration:** Maryam Sabzevari, Juho Rousu.

**Software:** Maryam Sabzevari.

**Supervision:** Merja Penttilä, Juho Rousu.

**Validation:** Maryam Sabzevari.

**Visualization:** Maryam Sabzevari.

**Writing – original draft:** Maryam Sabzevari, Juho Rousu.

**Writing – review & editing:** Maryam Sabzevari, Sandor Szedmak, Paula Jouhten, Juho Rousu.

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
