## [Decision Letter · Decision Letter 0]

25 Aug 2021

Dear Dr. sabzevari,

Thank you very much for submitting your manuscript "Strain design optimization using reinforcement Learning" for consideration at PLOS Computational Biology.

As with all papers reviewed by the journal, your manuscript was reviewed by members of the editorial board and by several independent reviewers. In light of the reviews (below this email), we would like to invite the resubmission of a significantly-revised version that takes into account the reviewers' comments.

The originality of your work was well received all around, but a major issue has been recognized relating to the practical application of your approach: you don't provide biological validation of the method and it seems that it could be very hard to implement in practice. In addition several reviewers commented on the lack of discussion of your results in the biological context. In order for the paper to be accepted in a future version it is critical that you address these points clearly. Of course we would like to see your reply (and corresponding changes) relating to the other points raised by the reviewers too. However the experimental feasibility of the method must be addressed, otherwise the manuscript fails to fit in the publication scope of this journal.

We cannot make any decision about publication until we have seen the revised manuscript and your response to the reviewers' comments. Your revised manuscript is also likely to be sent to reviewers for further evaluation.

Sincerely,

Pedro Mendes, PhD

Associate Editor

PLOS Computational Biology

Daniel Beard

Deputy Editor

PLOS Computational Biology

Reviewer's Responses to Questions

**Comments to the Authors:**

Reviewer #1: The authors recognise that strain optimization is a combinatorial search problem, and that RL is a useful approach in such cases (provided the data – real or simulated - are available). Of course in the hands of DeepMnd RL has had considerable success e.g. in learning to play Go, and so on. They (wisely) choose a model-free approach, not least since there are no directly predictive relations between the observables (i) concentration and (ii) flux (both are variables).

I set out thinking that this would be very exciting, but TBH it promised rather more than it delivered. I think the gushing tone of the abstract needs to be toned down to recognise the limitations more clearly: (i) only three somewhat unexciting fluxes were measured, (ii) it required FORTY iterations, which is a huge number for an experimental program, and (iii) many of the explanations were close to non-existent. It probably reflects the mathematical strengths of the first author, but these cannot come – in PlosBiol at least – at the expense of explaining what the findings actually meant. In consequence, it failed to convince me that I might seek to adopt their strategy, which is what I would wish it to have done. (Or alternatively that it was a nice idea but not in fact worth adopting in this way, a point nicely made in Harford T: How to

Make the World Add Up: ten rules for thinking differently about numbers. London: Bridge Stree Press, 2020.)

Abstract: “We demonstrate the method’s capabilities in comprehensive experiments using the genome-scale kinetic model of E. coli, k-ecoli457, as a surrogate for in vivo cell behaviour”. These were simulations, not experiments. Call them so. Comprehensive is a claim, not a fact.

I am not qualified other than to take the maths on pp 3-4 on trust. The background of the authors suggests this is reasonable.

The multi-agent aspect is nice as it recognises that this is effectively how experiments are performed in a DBTL cycle.

The authors might care to contrast their approach with BOED as in

Foster A, Jankowiak M, Bingham E, Horsfall P, Teh YW, Rainforth T, Goodman N: Variational Bayesian Optimal Experimental Design. arXiv 2019:1903.05480v05483.

Vanlier J, Tiemann CA, Hilbers PAJ, van Riel NAW: A Bayesian approach to targeted experiment design. Bioinformatics 2012; 28:1136-1142.

Fig 2. Numbers in heat map unreadable. Also try to choose non red-green for those who are colour blind. https://colorbrewer2.org/#type=diverging&scheme=BrBG&n=3 gives suggestions.

“As we observe in this figure, the algoritmh [sic] tunes the enzyme levels in generally smooth fashion while the larger changes correspond to replacing the worst agent with another”. Maybe so, but I cannot see this and so it needs a much better explanation. Tell me what I am supposed to look at and what I am supposed to draw from it.

I don’t understand the units in Figs 2 and 3 – you can’t have more moles out than went in…

Yield improvement p8. 40 iteration is not that realistic for real DBTL programs. It would be good to show the time course of the median per iteration.

Fig 3. Why is succinate different from ethanol and acetate?

Table 2. Strain stability measure. Needs far more explanation – what am I supposed to infer? Also I doubt that the second DP has much meaning, or even the first…

Fig 4. Again you tell us what you did but not what we are supposed to make of the results.

“The enzyme levels can be quantified using targeted mass spectrometry based proteomics approaches found useful in optimizing production”. I doubt it. Transcriptomics typically provide a much better surrogate (Machado D, Herrgård M: Systematic evaluation of methods for integration of transcriptomic data into constraint-based models of metabolism. PLoS Comput Biol 2014; 10:e1003580).

Reviewer #2: This is a good paper, the proposed reinforcement learning (RL) method is novel to the field of strain engineering, well explained and sound. The benchmark with other methods is also fair and clear. The introduction of noise is fully relevant especially where dealing with strain engineering. Yet, I find the method a bit limited in scope. First, one needs a strain ODE kinetics models and very few are available. Secondly, the method is applied only to a predefined set of enzymes and that information is not always available. Finally, to experimentally implement the method one needs to regulate the expression levels of enzymes (via promoters, RBSs..) and this is not a straightforward task where model simulations always agree with experimental results.

I would advise the authors to add in the discussion the current limitations of their method and discus if and how a similar method could be developed and used with genome scale models (GEMs) and steady state dynamics as there are plenty of such models, methods and results in the literature. The authors could also comment and if and how the actions could be simplified to propose set of genes to be knocked out. Note that in that latter case and when using GEMs, results could be compared with MILP solvers like OptKnock.

Minor comments:

Page 2: In addition to references 6-8 which make use of machine learning but not RL there are few papers explicitly making use of RL in the context of bioproduction and it is worth mentioning these //doi.org/10.1016/j.jprocont.2018.07.013 , //doi.org/10.1021/acssynbio.9b00447, //doi.org/10.1016/j.compchemeng.2019.106649 , //doi.org/10.1371/journal.pcbi.1007783 .

Figure 1. It would be wise to define the symbols used in the caption. Some can be found in the text but not all. Would also be wise to indicate where in the flow chart the policy is learned and used.

Page 4. The ML engine to build the policy is MMR (SVM based), could the author motivate this choice as other methods could have been used?

Page 5. The authors make use of a mixed centralized and decentralized training and defined groups where RL is carried out. Although within a group, exploration seems to be favored, I do not see the RL exploitation vs. exploration strategy being used nor discussed. Could the authors comment on that?

Table 1. The target reaction needs to be defined.

Figure. 2. The process is iterated 40 times but the Figure shows only 13 rows, what are they? Would also be good to state that the numbers and colors correspond to enzyme level.

Figure 4. The Figure shows yield decrement when noise is introduced separately to action, yield and state, has any test been carried out where noise was introduced simultaneously on the 3 elements?

In addition there are few typos which could be cleared with a spell checker.

Reviewer #3: This is an excellent paper outlining an important new approach to strain design optimization. I have two main areas of feedback:

1) The specific settings/decisions for tuning the MARL parameters are not justified. How was it decided for example the number of iterations and trajectories? Would slightly different options yield improved performance? Would one set of parameters be universally best for any problem, or would users of this method need to re-evaluate this for each new context? This should be explained.

2) Substantial editing is needed to improve the clarity of the manuscript. For example: on line 239 "algorithm" is spelled wrong and the word "a" is missing. The sentence starting on line 254 is grammatically incorrect.

**Have the authors made all data and (if applicable) computational code underlying the findings in their manuscript fully available?**

Reviewer #1: Yes

Reviewer #2: Yes

Reviewer #3: Yes

PLOS authors have the option to publish the peer review history of their article (what does this mean?). If published, this will include your full peer review and any attached files.

Reviewer #1: No

Reviewer #2: No

Reviewer #3: No
---

## [Decision Letter · Decision Letter 1]

26 Apr 2022

Dear Dr. sabzevari,

Thank you very much for submitting your manuscript "Strain design optimization using reinforcement Learning" for consideration at PLOS Computational Biology. As with all papers reviewed by the journal, your manuscript was reviewed by members of the editorial board and by several independent reviewers. The reviewers appreciated the attention to an important topic. Based on the reviews, we are likely to accept this manuscript for publication, providing that you modify the manuscript according to the review recommendations.

Sincerely,

Pedro Mendes, PhD

Associate Editor

PLOS Computational Biology

Daniel Beard

Deputy Editor

PLOS Computational Biology

[LINK]

Reviewer's Responses to Questions

**Comments to the Authors:**

Reviewer #2: I have read the revised version and agree with the modifications the authors have introduced. The authors have answered well all the points I raised with the exception of one.

The comment was “I would advise the authors to add in the discussion the current limitations of their method and discus if and how a similar method could be developed and used with genome scale models (GEMs) and steady state dynamics as there are plenty of such models, methods and results in the literature. The authors could also comment and if and how the actions could be simplified to propose set of genes to be knocked out. Note that in that latter case and when using GEMs, results could be compared with MILP solvers like OptKnock.“

As a reply to this comment the authors are presenting the difficulties and solutions to modulate enzyme expression level (a point that was raised earlier in the initial review). Could the above comment be addressed?

Reviewer #3: Thank you for your updates, this addresses my concerns and comments.

**Have the authors made all data and (if applicable) computational code underlying the findings in their manuscript fully available?**

Reviewer #2: Yes

Reviewer #3: Yes

PLOS authors have the option to publish the peer review history of their article (what does this mean?). If published, this will include your full peer review and any attached files.

Reviewer #2: No

Reviewer #3: No

Figure Files:

Data Requirements:

Reproducibility:

References:

---

## [Editor Report · Decision Letter 2]

6 May 2022

Dear Dr. sabzevari,

We are pleased to inform you that your manuscript 'Strain design optimization using reinforcement Learning' has been provisionally accepted for publication in PLOS Computational Biology.

Best regards,

Pedro Mendes, PhD

Associate Editor

PLOS Computational Biology

Daniel Beard

Deputy Editor

PLOS Computational Biology

---

## [Editor Report · Acceptance letter]

26 May 2022

PCOMPBIOL-D-21-01222R2 

Strain design optimization using reinforcement learning

Dear Dr Sabzevari,

I am pleased to inform you that your manuscript has been formally accepted for publication in PLOS Computational Biology. Your manuscript is now with our production department and you will be notified of the publication date in due course.

With kind regards,

Anita Estes
